# Uncertainty-driven regulation of learning and exploration in adolescents: A computational account

**Marieke Jepma** *, **Jessica V. Schaaf, Ingmar Visser, Hilde M. Huizenga**

Department of Psychology, University of Amsterdam, Amsterdam, the Netherlands

* m.jepma@uva.nl

**Data Availability Statement:** All single-trial behavioural/task data are available from the OSF database (https://osf.io/wvqsj/)

## Abstract

Healthy adults flexibly adapt their learning strategies to ongoing changes in uncertainty, a key feature of adaptive behaviour. However, the developmental trajectory of this ability is yet unknown, as developmental studies have not incorporated trial-to-trial variation in uncertainty in their analyses or models. To address this issue, we compared adolescents' and adults' trial-to-trial dynamics of uncertainty, learning rate, and exploration in two tasks that assess learning in noisy but otherwise stable environments. In an estimation task—which provides direct indices of trial-specific learning rate—both age groups reduced their learning rate over time, as self-reported uncertainty decreased. Accordingly, the estimation data in both groups was better explained by a Bayesian model with dynamic learning rate (Kalman filter) than by conventional reinforcement-learning models. Furthermore, adolescents' learning rates asymptoted at a higher level, reflecting an over-weighting of the most recent outcome, and the estimated Kalman-filter parameters suggested that this was due to an overestimation of environmental volatility. In a choice task, both age groups became more likely to choose the higher-valued option over time, but this increase in choice accuracy was smaller in the adolescents. In contrast to the estimation task, we found no evidence for a Bayesian expectation-updating process in the choice task, suggesting that estimation and choice tasks engage different learning processes. However, our modeling results of the choice task suggested that both age groups reduced their degree of exploration over time, and that the adolescents explored overall more than the adults. Finally, age-related differences in exploration parameters from fits to the choice data were mediated by participants' volatility parameter from fits to the estimation data. Together, these results suggest that adolescents overestimate the rate of environmental change, resulting in elevated learning rates and increased exploration, which may help understand developmental changes in learning and decision-making.

## Author summary

To successfully learn the value of stimuli and actions, people should take into account their current (un)certainty about these values: Learning rates and exploration should be

**Funding:** This work was funded by the Dutch National Science Foundation, NWO, (VICI 453-12-005). The funder had no role in study design, data collection and analysis, decision to publish, or preparation of the manuscript.

**Competing interests:** The authors have declared that no competing interests exist.

high when one's value estimates are highly uncertain (in the beginning of learning), and decrease over time as evidence accumulates and uncertainty decreases. Recent studies have shown that healthy adults flexibly adapt their learning strategies based on ongoing changes in uncertainty, consistent with normative learning. However, the development of this ability prior to adulthood is yet unknown, as developmental learning studies have not considered trial-to-trial changes in uncertainty. Here, we show that adolescents, as compared to adults, showed a smaller decrease in both learning rate and exploration over time. Computational modeling revealed that both of these effects were due to adolescents overestimating the amount of environmental volatility, which made them more sensitive to recent relative to older evidence. The overestimation of volatility during adolescence may represent the rapidly changing environmental demands during this developmental period, and can help understand the surge in real-life risk taking and exploratory behaviours characteristic of adolescents.

## Introduction

Learning to predict future events based on past experiences, and choosing one's actions accordingly, is essential for adaptive behaviour. Across the lifespan, experience-driven learning is used to accomplish a broad range of behaviours, such as riding a bike, functioning adaptively in different social contexts, and discovering your favorite restaurants. While the types of behaviours and events that we learn about obviously change with age, recent studies suggest that the learning process itself also undergoes substantial developmental changes [1–5]. More insight in the nature of these developmental changes can help optimize educational strategies and policy interventions aimed at specific age groups.

To uncover the neurocognitive processes that give rise to developmental changes in experience-based learning, computational models of reinforcement learning are becoming an increasingly popular tool [1]. These models describe learning as the iterative updating of expectations in response to discrepancies between expected and observed events, i.e., prediction errors [6, 7]. Importantly, how much expectations are updated following each prediction error depends on the *learning rate*, such that higher learning rates result in a stronger influence of recent compared to more historical events. In addition, when applied to choice data, these models include a decision function that translates expected values into choice probabilities. The probability that the model selects the option with the highest expected valued is often controlled by the *inverse-temperature* parameter, such that a higher inverse temperature increases the likelihood that the model will choose the option with the highest expected value. Thus, a lower inverse temperature can account for a more random, or exploratory, choice strategy. However, the inverse temperature also accounts for other sources of variance in choice behaviour that cannot be explained by the model (e.g., due to model misspecifications), which is important to consider when interpreting differences in inverse-temperature estimates between groups or conditions.

Recent developmental studies that used computational models have provided evidence that age-related differences in reinforcement-learning performance—usually defined as choice accuracy in multi-armed bandit tasks—are related to developmental changes in learning rate [3, 8–10], sensitivity to prediction errors [10, 11], and exploratory choice behaviour (inverse temperature) [9, 12]. However, a crucial aspect of reinforcement learning that remains to be addressed in the developmental field is the regulation of learning strategies as a function of ongoing changes in uncertainty. Specifically, the optimal rates of learning and exploration depend on various sources of uncertainty, including imperfect knowledge about the current

state of the environment (estimation uncertainty, or ambiguity), the amount of random outcome variability (noise, or risk), and the degree of environmental change (volatility) [13–15]. For example, largely unknown or rapidly changing environments warrant high learning rates to facilitate learning, whereas noisy but otherwise stable and well-known environments warrant low learning rates to prevent excessive expectation updating [16]. Similarly, exploration is beneficial when uncertainty is high, whereas exploitation of the option with the highest expected value is the best strategy when uncertainty is low [17].

Studies in adults that used (Bayesian) models with a dynamic learning rate or dynamic exploration parameter have provided evidence that healthy adults reduce their learning rate [18–20] and degree of exploration [21, 22] over the course of stable task periods—as estimation uncertainty decreases—consistent with normative behaviour. However, developmental reinforcement-learning studies have not incorporated uncertainty in their analyses and models but so far assumed constant rates of learning and exploration, likely providing an incomplete description of the underlying learning and decision processes. Thus, it is currently unclear whether age-related changes in reinforcement learning performance reflect changes in (i) learning and/or exploration per se and/or (ii) their adaptive, uncertainty-driven, regulation over time.

The aim of the present study was to shed more light on potential developmental changes in uncertainty-driven regulation of learning and exploration. We focused on the period of adolescence, as previous studies suggest that tolerance for uncertainty peaks during this period [23–25]. Specifically, we compared trial-to-trial dynamics of adolescents' and adults' uncertainty, learning rate, and choice accuracy, during two noisy but otherwise stable reinforcement-learning tasks. Furthermore, we specified the latent processes underlying age-related differences in the observed dynamics using computational models. In the first task we examined age-related differences in the uncertainty-driven regulation of learning rate, unconfounded by potential differences in exploration. In this task, participants repeatedly estimated the mean outcome generated by a noisy process, and reported how certain they were about each estimate. Because the mean outcome can be estimated increasingly precise as the number of observed outcomes increases, optimal performance warrants a decrease in learning rate over time. In the second task, participants made repeated choices between two options. Each option generated outcomes according to a noisy process with a different mean. Optimal performance in this task warrants not only a decrease in learning rate, but also a decrease in exploration over time, and we examined age-related differences in the adaptation of both variables.

To foreshadow the results, both the adults and adolescents decreased their learning rate (in the first task) and their degree of exploration (in the second task) over time, consistent with normative behaviour. However, the adolescents showed higher asymptotic learning rates and more persistent exploration. Computational modeling analyses suggest that these effects are due to adolescents overestimating the volatility of the task environment.

## Results

### Preregistration

We preregistered both tasks (https://aspredicted.org/blind.php?x=fi89q7 and http://aspredicted.org/blind.php?x=av4td4 for the estimation and choice task, respectively). We also preregistered the reported analyses of the estimation data, but not of the choice data. Deviations from the preregistered analyses are described and explained in S1 Text.

### Experimental design

Participants (35 adults and 25 adolescents, matched with regard to educational level) performed two reinforcement-learning tasks in which successful performance requires the

integration of sequentially observed outcomes: An estimation task and a choice task, respectively (see Methods for details). In the estimation task, participants observed a sequence of outcomes (numbers) drawn from a Gaussian distribution. The mean of this distribution was fixed within each block but varied across blocks, and the standard deviation (SD) of the distribution was 4 or 8, in five blocks each (Fig 1A). Before each new outcome was revealed, participants estimated the average outcome in that block, and indicated how certain they were about this estimate (Fig 1B). This task measures the expectation-updating process that is the basis of reinforcement-learning models, and provides direct, trial-specific indices of prediction error (the observed minus the estimated outcome) and learning rate (the change in a participant's estimate from one trial to the next, divided by the most recent prediction error[19, 20]). In the choice task (a two-armed bandit task; Fig 1C), the same participants made repeated choices between two options that generated outcomes (points) from two different Gaussian distributions (the means differed 10 or 20 points, in four blocks each; the SD was always 8), with the aim to obtain as many points as possible. Each block consisted of 20 trials, and participants learned about new choice options from scratch during each block. Besides estimating the mean outcome associated with each option, this task involves the tradeoff between choosing the option that is currently expected to be the best (exploitation) and gathering more information about the other option (exploration). Except for the specific means and SDs used, participants were fully informed about the task structure and procedure before starting each task. Also, the start of each new block was clearly indicated, by means of a 'new-block' screen.

Below, we first report a series of multilevel regression and mediation analyses that examine potential differences between adolescents and adults in (i) trial-to-trial variation in certainty

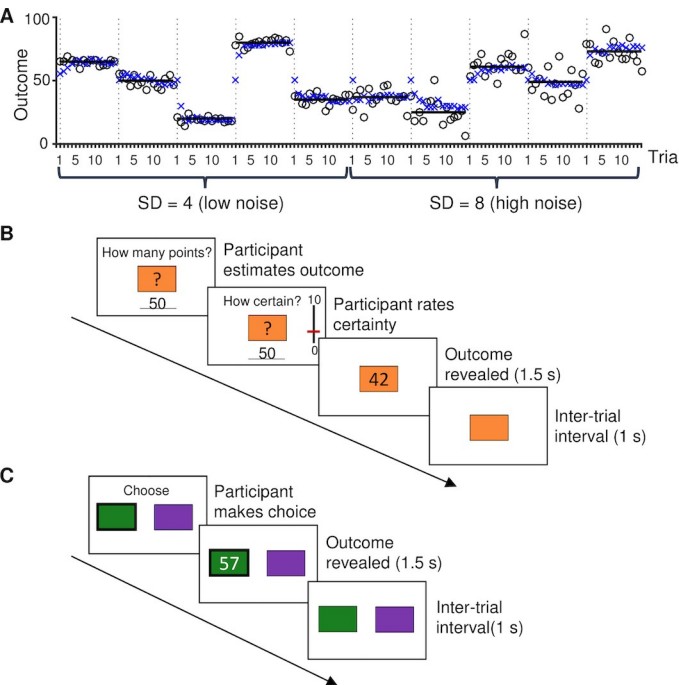

**Fig 1. Experimental tasks. A**. Example of an adult participant's estimated and observed outcomes during the estimation task. This participant completed the low-noise (SD = 4) followed by the high-noise (SD = 8) blocks. The dotted vertical lines indicate the first trial of each block. The circles indicate the outcomes on each trial; these were drawn from a normal distribution which mean (horizontal black line) was constant during each 15-trial block. The blue crosses indicate the participant's estimates. **B.** One trial of the estimation task (participants completed 10 blocks of 15 trials each). **C.** One trial of the choice task (participants completed 8 blocks of 20 trials each).

ratings and learning rate during the estimation task; and (ii) trial-to-trial variation in choice accuracy during the choice task. Then, to further specify the nature of the observed age effects, we apply normative learning models that quantify the latent processes thought to underlie the adaptation of learning rate and exploration over time. Finally, we report relationships between model parameters governing the adaptation of learning rate and exploration across participants, to examine evidence for a shared mechanism underlying developmental changes in these two processes.

## Self-reported certainty increases over time, similarly in adolescents and adults

As expected, participants' certainty ratings in the estimation task increased with the number of observed outcomes. It did so in a sublinear way (linear and quadratic effect of trial, $t(58) = 5.8$, $p < .001$ and $t(58) = 3.9$, $p < .001$, respectively; Fig 2A). In addition, certainty was higher in the low-noise than the high-noise blocks ($t(58) = 3.1$, $p = .003$), and the effect of noise increased linearly over trials (noise x trial-linear interaction, $t(58) = 4.2$, $p < .001$).

Overall certainty did not differ between the adults and adolescents ($t(58) = .07$, $p = .94$). The two age groups showed similar increases in certainty over time (age group x trial-linear and age group x trial-quadratic interaction, $t(58) = .60$, $p = .55$ and $t(58) = 1.6$, $p = .12$, respectively), and did not differ with regard to the effect of noise level on certainty either (age group x noise interaction, $t(58) = .29$, $p = .77$). The only age-related effect on certainty was an age group x noise x trial-linear interaction ($t(58) = 3.1$, $p = .003$), reflecting that the increase in the effect of noise level over trials was stronger in the adolescents.

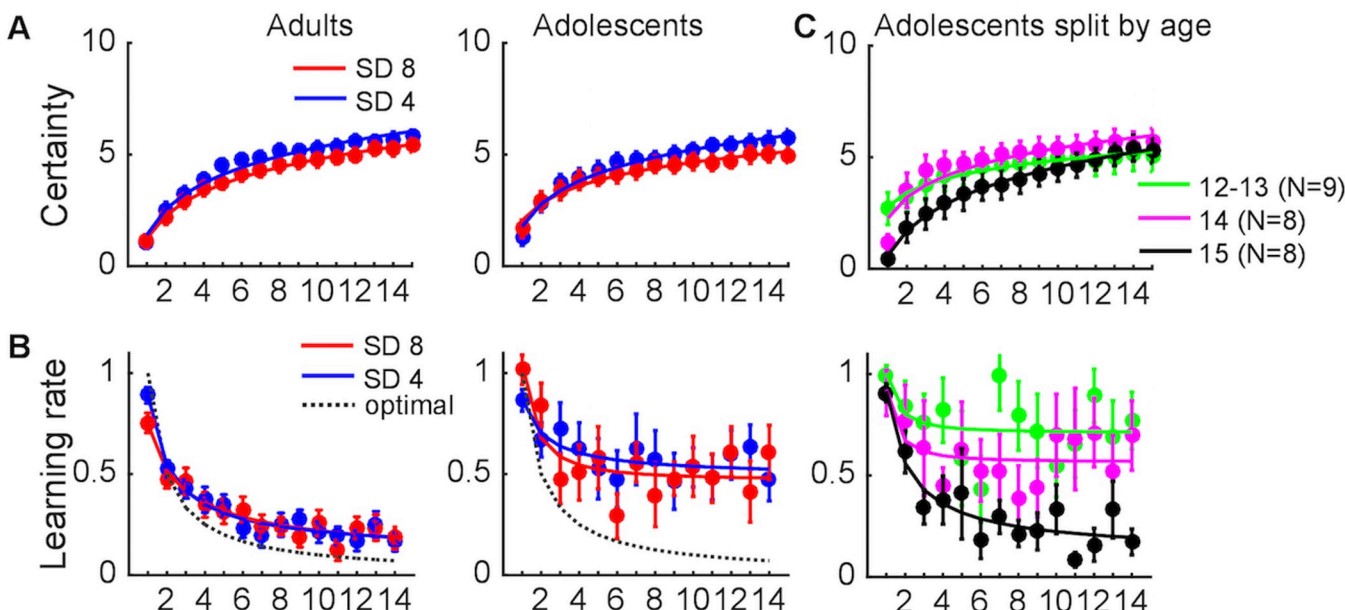

**Fig 2. Changes in certainty and learning rate over time. A**. Certainty rating as a function of trial, noise level and age group. **B**. Learning rate (directly computed from the estimation data) as a function of trial, noise level and age group. Note that we could not compute learning rate for the last trial of each block (trial 15) as participants did not update their estimate after the fifteenth observation. The dotted gray lines show the optimal learning rate on each trial of this task. **C**. Certainty rating and learning rate in the adolescents, as a function of trial and age. Error bars indicate standard errors. The lines are power-function fits to the mean time-courses of each age group.

## Adolescents show a smaller decrease in learning rate over time than adults

Consistent with normative learning, participants' learning rate in the estimation task (directly computed from the estimation data, see Methods) decreased as more outcomes were observed, in a sublinear way (linear and quadratic effect of trial, $t(58) = 2.5$, $p = .02$ and $t(58) = 2.6$, $p = .01$, respectively; Fig 2B). Learning rate was not affected by noise level ($t(58) = .85$, $p = .40$), and there were no noise x trial interactions (noise x trial-linear and noise x trial-quadratic interaction, $t(58) = .14$, $p = .89$ and $t(58) = 2.0$, $p = .055$, respectively).

The dotted gray lines in Fig 2B show the optimal learning rate on each trial of the estimation task (see S2 Text for an explanation). Although both groups used higher-than-optimal learning rates after the first few trials—indicating that their estimates were driven too much by the most recent outcomes—this deviation from optimality was stronger in the adolescents. The adolescents used overall higher learning rates than the adults ($t(58) = 3.2$, $p = .002$). In addition, the adolescents showed a smaller decrease in learning rate over time (age group x trial-linear interaction, $t(58) = 2.3$, $p = .02$; age group x trial-quadratic interaction, and $t(58) = 1.4$, $p = .16$). The effect of noise level on learning rate did not differ between the two groups (age group x noise interaction, $t(58) = .52$, $p = .61$). Finally, there was an age group x noise x trial-quadratic interaction ($t(58) = 2.1$, $p = .04$), reflecting a relatively stronger quadratic trial effect in the high- than the low-noise blocks in the adolescents compared to the adults.

## Learning rate decreases over the course of early adolescence

We also explored age-related differences within the group of adolescents (Fig 2C), by repeating the previous analyses on certainty and learning rate for the group of adolescents only, using age—which varied between 12 and 15—as a continuous second-level regressor.

The analysis on certainty revealed no main effect of adolescent age ($t(23) = .61$, $p = .55$), but a trend for a stronger linear increase in certainty over trials with increasing age (age x trial-linear interaction, $t(23) = 1.9$, $p = .07$). There were no age x noise ($t(23) = .72$, $p = .48$) or age x trial x noise interactions on certainty ($t(23) = 1.0$, $p = .33$ and $t(23) = .96$, $p = .35$ for the age x trial-linear and age x trial-quadratic interaction, respectively).

The analysis on learning rate revealed that overall learning rate decreased with increasing adolescent age (main effect of age, $t(23) = 3.5$, $p = .002$), but there were no age x trial interactions (age x trial-linear and age x trial-quadratic interaction, $t(23) = 1.0$, $p = .33$ and $t(23) = 1.3$, $p = .21$, respectively). Fig 2C suggests that learning rates did not differ between the adults and the 15-year-olds, but were elevated in the 12-14-year-olds. To test this idea, we divided the adolescents in three age groups—12/13-year-olds (combined because of the low number of adolescents of each age), 14-year-olds, and 15-year-olds—and compared each of these groups to the adults. These analyses revealed that the 12/13-year-olds used a higher overall learning rate than the adults (main effect of age, $t(42) = 3.1$, $p = .003$), and also reduced their learning rate less over time (age x trial-linear and age x trial-quadratic interaction, $t(42) = 2.1$, $p = .04$ and $t(42) = 3.9$, $p < .001$, respectively). The 14-year-olds used a higher overall learning rate than the adults (main effect of age, $t(41) = 2.4$, $p = .02$), but did not differ from the adults in the adjustment of learning rate over time (age x trial-linear and age x trial-quadratic interaction, $t(41) = 1.7$, $p = .10$ and $t(41) = 1.4$, $p = .17$, respectively). Finally, the 15-year-olds did not differ from the adults in either overall learning rate (main effect of age, $t(41) = .02$, $p = .98$) or its adjustment over time (age x trial-linear and age x trial-quadratic interaction, $t(41) = 1.0$, $p = .32$ and $t(41) = 1.2$, $p = .24$, respectively). Although these results should be taken with caution given the small number of adolescents of each age, they provide preliminary evidence that the ability to regulate learning rate emerges over the course of early adolescence, and is matured around the age of fifteen.

### Self-reported certainty predicts learning rate, similarly in adolescents and adults

We next examined whether self-reported certainty is predictive of learning rate when controlling for the number of trials that have been observed already, and whether self-reported certainty mediates the decrease in learning rate over trials, using multilevel mediation (Fig 3A). To examine potential differences between adults and adolescents, we included age group as a second-level moderator in the mediation model.

Certainty increased over trials, as reported above (path $a$, $p < .001$). Importantly, higher certainty predicted lower learning rates when controlling for trial (path $b$, $p < .001$; see Fig 3B for the relationship between certainty and learning rate, not controlled for trial). In addition, certainty formally mediated the effect of trial number on learning rate ($a^*b$, $p < .001$). When controlled for certainty, the relationship between trial number and learning rate remained significant (path $c'$, $p < .001$), implying a partial mediation. Age group moderated the total effect of trial on learning rate (path $c$; $p = .04$), consistent with the previous analysis on learning rate. However, age group did not moderate any of the other paths of the mediation model (all $p$'s > .14), suggesting that the adolescents and adults similarly adjusted their learning rate based on their current certainty.

### Adolescents show a smaller increase in choice accuracy over time than adults

In the choice task, participants' choice accuracy (proportion of choices for the option associated with the highest outcome-generating distribution) showed a sublinear increase over time (linear and quadratic effects of trial number, $z = 7.6$, $p < .001$ and $z = 2.2$, $p = .03$, respectively; Fig 4). Overall choice accuracy was higher in the adults than the adolescents ($z = 4.1$, $p < .001$). Furthermore, the adults showed a faster increase in choice accuracy over trials (age group x trial-linear and age group x trial-quadratic interactions, $z = 5.1$, $p < .001$ and $z = 2.4$, $p = .02$, respectively).

To examine initial evidence for changes in exploration over time (see the next section for model-based analyses), we classified each choice as exploitative or exploratory in the following way. For each trial, we computed the average outcome that had been observed for each option

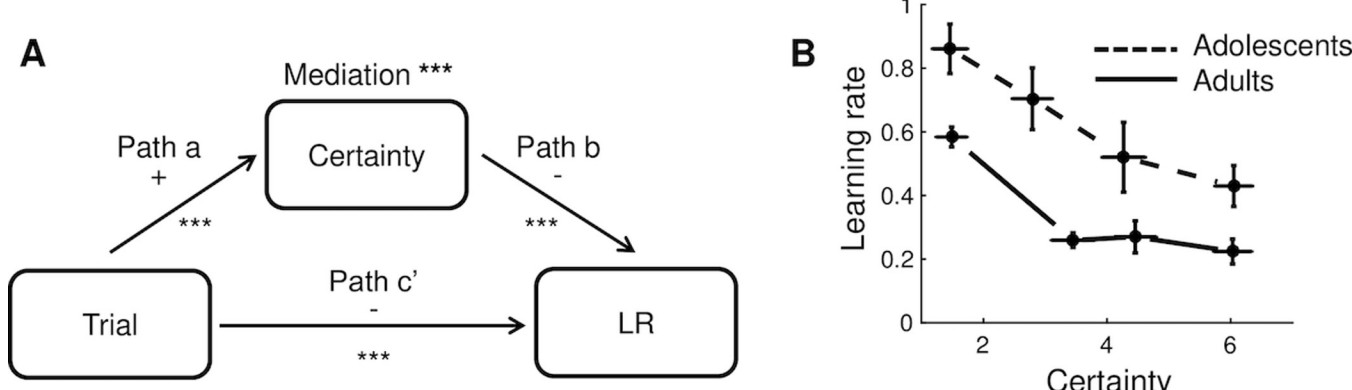

**Fig 3. Relationships between trial number, certainty rating and learning rate during the estimation task. A**. Mediation model and results. LR = learning rate. *** $p < .001$. **B**. Learning rate as a function of certainty rating and age group. We divided each participant's learning rates into four bins according to their trial-specific certainty ratings, and plotted the group-average data for each bin, separately for each age group. Note that we binned the data for plotting purposes, but used single-trial measures in our statistical analyses. Error bars indicate standard errors.

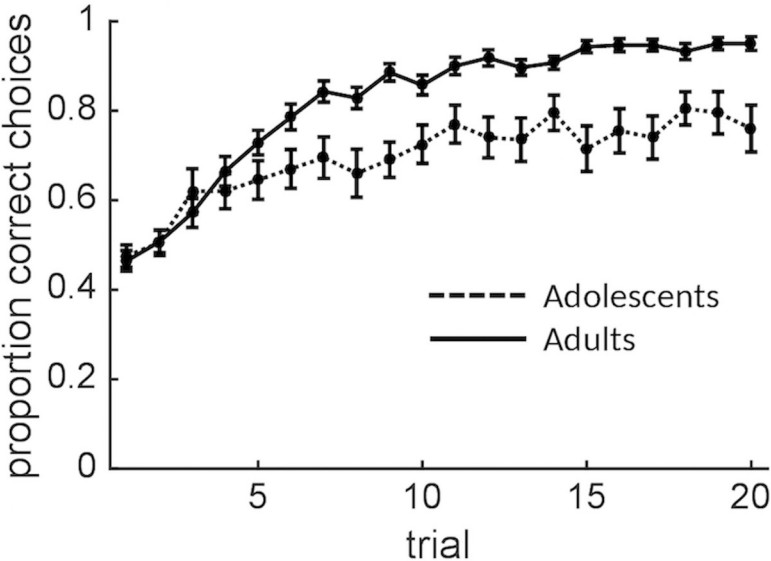

**Fig 4. Choice accuracy as a function of trial and age group.** Error bars indicate standard errors.

so far (a model-free estimate of expected value), and we classified choices for the option with the higher and lower expected value as, respectively, exploitative and exploratory (only including the trials at which both options had been chosen at least once before). We then tested for main and interaction effects of trial (linear and quadratic) and age group on exploratory choices, as defined above. Importantly, to control for changes in relative expected value over time, we also modeled the absolute difference between the two options' expected value. As expected, exploration decreased as the difference in expected value increased ($z = 5.0$, $p < .001$). This effect of relative expected value on exploration was larger in the adults than the adolescents ($z = 4.9$, $p < .001$). This analysis also revealed a linear decrease in exploration over trials ($z = 5.8$, $p < .001$). Finally, the adolescents, as compared to the adults, showed more exploration overall ($z = 4.7$, $p < .001$) as well as a smaller linear decrease in exploration over trials ($z = 4.3$, $p < .001$). These findings provide initial evidence that participants—and adults more so than adolescents—reduced their degree of exploration over time, consistent with normative behaviour on this task.

## Computational models

The behavioural results from the two tasks suggest that (i) self-reported uncertainty, and its effect on learning rate, do not differ between adolescents and adults (estimation task); but (ii) adolescents show a smaller decrease in learning rate over time, resulting in higher asymptotic learning rates (estimation task); and (iii) adolescents show more, and more sustained, exploration than adults, resulting in lower asymptotic choice accuracy (choice task). To formalize and quantify the latent processes that may underlie these effects, we applied a series of computational models to the estimation and choice data.

We modeled the expectation-updating process in both tasks using (i) a standard reinforcement-learning model (Model 1) with one constant learning rate; (ii) an asymmetric reinforcement-learning model (Model 2) with two separate constant learning rates for positive and negative prediction errors; (iii) a Kalman filter (Model 3), which describes learning in terms of optimal statistical (Bayesian) inference, in which learning rate varies as a function of

estimation uncertainty; and (iv) a hybrid Pearce-Hall model (Model 4), in which the learning rate varies over time in a non-Bayesian way. For fits to the choice data, we combined each of these learning models with a softmax function, which translates the options' expected values into choice probabilities. We used two versions of this function: (i) one in which the inverse temperature is constant over trials (Model 1A-4A); and (ii) one in which the inverse temperature varies over trials (Model 1B-4B) [22]. Full descriptions of the models, and their equations, can be found in the Methods; here we provide a brief overview of their main principles and parameters.

**Model 1: standard reinforcement-learning model.** This model updates an option's expected (mean) outcome following each new observation in proportion to the prediction error (observed minus expected outcome), according to a standard reinforcement-learning algorithm (delta rule [7]). Learning rate parameter $\alpha$, which is constant over trials, controls the degree of expectation updating, such that higher values of $\alpha$ result in stronger updating towards the most recent outcome. This model has one free parameter ($\alpha$).

**Model 2: asymmetric reinforcement-learning model.** This model is identical to Model 1, except that it uses two separate learning rates to control expectation updating following positive and negative prediction errors: $\alpha_+$ and $\alpha_-$, respectively. Thus, this model has two free parameters. This model may explain age-related differences in choice behaviour in terms of differential sensitivity to better- and worse-than-expected outcomes [3, 9, 10]. Note that a learning asymmetry is less likely in the estimation task, as positive and negative prediction errors in this task do not reflect better- and worse-than-expected outcomes, respectively. However, for completeness, we applied all models to both tasks.

**Model 3: Kalman filter.** This model considers learning as Bayesian inference, tracking estimates of an option's mean (expected) outcome, as well as the precision of those estimates [26, 27]. In contrast to the reinforcement-learning model, the effective learning rate on each trial depends on the precision of the expectation at the onset of that trial, such that less precise expectations lead to greater updating towards the most recent outcome. In addition, the Kalman filter assumes that the mean outcome varies over time according to a Gaussian random walk process, adding some uncertainty to the expectation after each trial. This model has three parameters characterizing the participant's belief about the outcome-generating process: the variance of the random walk process (drift variance; $\sigma_\eta^2$), the variance in outcomes around the mean on any given trial (noise variance; $\sigma_\varepsilon^2$), and the uncertainty of the initial expectation (initial prior variance; $s_1^2$). These can be considered indices of assumed volatility, assumed noise level, and initial estimation uncertainty, respectively. Only the ratios among these three parameters matter; hence we fixed $\sigma_\varepsilon^2$ to 1. Thus, this model has two free parameters, $s_1^2$ and $\sigma_\eta^2$ (both estimated relative to $\sigma_\varepsilon^2$), which determine the initial and asymptotic learning rate, respectively.

**Model 4: reinforcement learning/Pearce-Hall hybrid model.** This model also has a dynamic learning rate, but the learning rate on each trial is defined in a non-Bayesian way, by scaling it by a dynamic 'associability' term which reflects the unexpectedness of previous outcomes [28]. Specifically, the associability on a given trial, $\alpha_t$, is a weighted average of the associability and the absolute prediction error on the previous trial. Decay parameter $\eta$ determines the relative weights of these two terms. This model can be seen as a hybrid between the standard reinforcement learning model and the Pearce-Hall associability model [29–31]. It has three free parameters: the initial associability $\alpha_1$, decay parameter $\eta$, and the constant component of the learning-rate $\kappa$ (the learning rate on trial $t$ is the product of $\kappa$ and $\alpha_t$).

**Constant and dynamic softmax function.** To convert expectations into choice probabilities (for fits to the choice data), we combined each learning model with a constant (Model 1A,

2A, 3A and 4A) and a dynamic (Model 1B, 2B, 3B and 4B) softmax function. The softmax function computes the probability that each option is chosen based on the difference in expected outcome between the two options, and inverse-temperature parameter $\beta$ indicates how strongly choice probabilities are linked to the difference in expected outcomes: A higher inverse temperature could reflect a stronger exploitation of the option with the highest expected value, or a better ability of the model to capture participants' choices. In Models 1A-4A (constant softmax), the inverse temperature is constant over trials. In Models 1B-4B (dynamic softmax), we allowed the inverse temperature to change over trials according to a power function, with the direction and rate of change controlled by parameter $c$: Positive and negative values of $c$ produce an increasing and decreasing inverse temperature over trials, respectively, and larger absolute values of $c$ result in faster change. The free parameter $\theta$ determines the inverse temperature on trial 10 (halfway a task block). Thus, the constant softmax function has one free parameter ($\beta$), and the dynamic softmax function has two free parameters ($c$ and $\theta$).

## Model comparison

We applied Model 1–4 to participants' trial-to-trial estimates during the estimation task (separately for the low- and high-noise blocks), and Models 1A-4A and 1B-4B to participants' trial-to-trial choices during the choice task. All models were fitted separately to the adolescent and adult data, using a hierarchical Bayesian approach (see Methods). We compared the models' performance using the deviance information criterion (DIC), a hierarchical modeling generalization of the AIC which is easily computed in hierarchical Bayesian model-selection problems using Markov chain Monte Carlo (MCMC) sampling [32] (Table 1). In addition, we transformed the DIC values to a probability scale (model weights), enabling a more intuitive comparison of the probabilities of each model being the best model, given the data and the set of candidate models [33].

**Table 1. DIC values and (in parentheses) corresponding model weights for fits of each model to each data set.**

|  | Estimation task | | | |
|---|---|---|---|---|
|  | Low noise | | High noise | |
|  | adults | adolescents | adults | adolescents |
| Model1: RL | 13,769 ($<$ .01) | 11,125 ($<$ .01) | 15,554 ($<$ .01) | 12,447 ($<$ .01) |
| Model 2: RL2 | 13,679 ($<$ .01) | 11,114 ($<$ .01) | 15,506 ($<$ .01) | 12,408 ($<$ .01) |
| Model 3: KF | **10,864 ($>$ .99)** | **9,949 ($>$ .99)** | **12,608 ($>$ .99)** | **11,511 ($>$ .99)** |
| Model 4: PH | 10,951 ($<$ .01) | 10,042 ($<$ .01) | 12,831 ($<$ .01) | 11,584 ($<$ .01) |

|  | Choice task | |
|---|---|---|
|  | adults | adolescents |
| Model 1A: RL/constant $\beta$ | 3,992 ($<$ .01) | 4,066 ($<$ .01) |
| Model 1B: RL/dynamic $\beta$ | 3,596 ($<$ .01) | 3,904 ($<$ .01) |
| Model 2A: RL2/constant $\beta$ | 3,932 ($<$ .01) | 3,996 ($<$ .01) |
| Model 2B: RL2/dynamic $\beta$ | 3,587 ($<$ .01) | **3,866 ($>$ .99)** |
| Model 3A: KF/constant $\beta$ | 3,865 ($<$ .01) | 4,004 ($<$ .01) |
| Model 3B: KF/dynamic $\beta$ | 3,548 ($<$ .01) | 3,904 ($<$ .01) |
| Model 4A: PH/constant $\beta$ | 3,962 ($<$ .01) | 4,066 ($<$ .01) |
| Model 4B: PH/dynamic $\beta$ | **3,513 ($>$ .99)** | 3,904 ($<$ .01) |

Notes: The values of the best-fitting models (lowest DIC) are shown in bold. RL, RL2, KF, and PH are the reinforcement-learning, asymmetric reinforcement-learning, Kalman-filter, and reinforcement learning/Pearce-Hall hybrid model, respectively, and $\beta$ = inverse temperature.

For fits to the estimation data, the models with dynamic learning rates (models 3 and 4) clearly outperformed the models with constant learning rates (models 1 and 2) for both age groups and noise levels, corroborating the behavioural learning-rate data. Furthermore, Model 3 (Kalman filter) outperformed Model 4 (reinforcement learning/Pearce-Hall hybrid model) for both age groups and noise levels. This suggests that both the adolescents and adults used Bayesian inference to update their estimates.

For fits to the choice data, model 4B (the reinforcement learning/Pearce-Hall hybrid model + dynamic softmax) performed best for the adults, while model 2B (asymmetric reinforcement-learning model + dynamic softmax) performed best for the adolescents. This suggests that when making repeated choices between two options—as compared to estimating the average value of one stimulus—adults use a simpler, non-Bayesian, mechanism to adjust their learning rate over time, while adolescents do not adjust their learning rate over time. Importantly, all learning models performed better when combined with a dynamic than a constant softmax function, for both age groups, suggesting that both the adults and adolescents did adjust their degree of exploration over time, or that the models' ability to capture participants' choices changed as learning progressed.

For fits to the choice data, we also computed the probability of each choice under each of the models—given the posterior medians of the individual-level parameters and the outcomes observed up to that choice—as an intuitive index of how well the models explain the choice data. The average probability of the observed choices (averaged across trials and participants) under the best fitting model was .78 for the adults, and .65 for the adolescents, consistent with the lower inverse temperature for the adolescents. The probabilities of the observed choices under the other models ranged from .56 to .78 for the adults, and from .54 to .65 for the adolescents (S1 Table).

## Parameter estimates

We next examined the parameter estimates of the best fitting models. We separately estimated group-level parameter distributions for the adolescent and adult participants, and denote the hyperparameters governing the central tendencies of the group-level distributions with overbars (e.g., $\overline{\theta}$ refers to the group-level mean of $\theta$). Fig 5A (left panels), C, and D (left panels) show the posterior distribution of the central tendency of each parameter's group-level distribution, for fits to the estimation and choice data, respectively. The corresponding medians and 95% highest-density intervals (HDIs) are reported in S2 Table. In the Kalman filter, $\sigma_\eta^2$ is the assumed variance of the random walk process (random variance in the mean outcome across trials) and $s_1^2$ is the initial prior variance (both estimated relative to the assumed noise variance, $\sigma_\varepsilon^2$, which was fixed at 1 to eliminate redundancy in model parameters). In the reinforcement learning/Pearce-Hall hybrid model, $\alpha_1$ is the initial associability, $\eta$ is a decay parameter that controls the dynamics of associability over time, and $\kappa$ is a fixed proportion that scales the associability on each trial (i.e., the learning rate on trial $t$ is the product of $\kappa$ and $\alpha_t$). In the asymmetric reinforcement learning model, $\alpha_+$ and $\alpha_-$ are the learning rates following positive and negative prediction errors, respectively. In the dynamic softmax function, $\overline{\theta}$ is the inverse temperature on trial 10 and $c$ controls the change in inverse temperature over trials (positive values cause an increase over trials).

**Estimation task.** For fits to the estimation data, the posterior distributions of $\overline{\sigma_\eta^2}$ (drift variance) for the adult participants essentially peaked at 0, its lower bound, suggesting that the adults correctly assumed a stationary outcome-generating process. The posterior distributions of $\overline{\sigma_\eta^2}$ for the adolescent participants lay close to 0 as well, but were considerably higher than

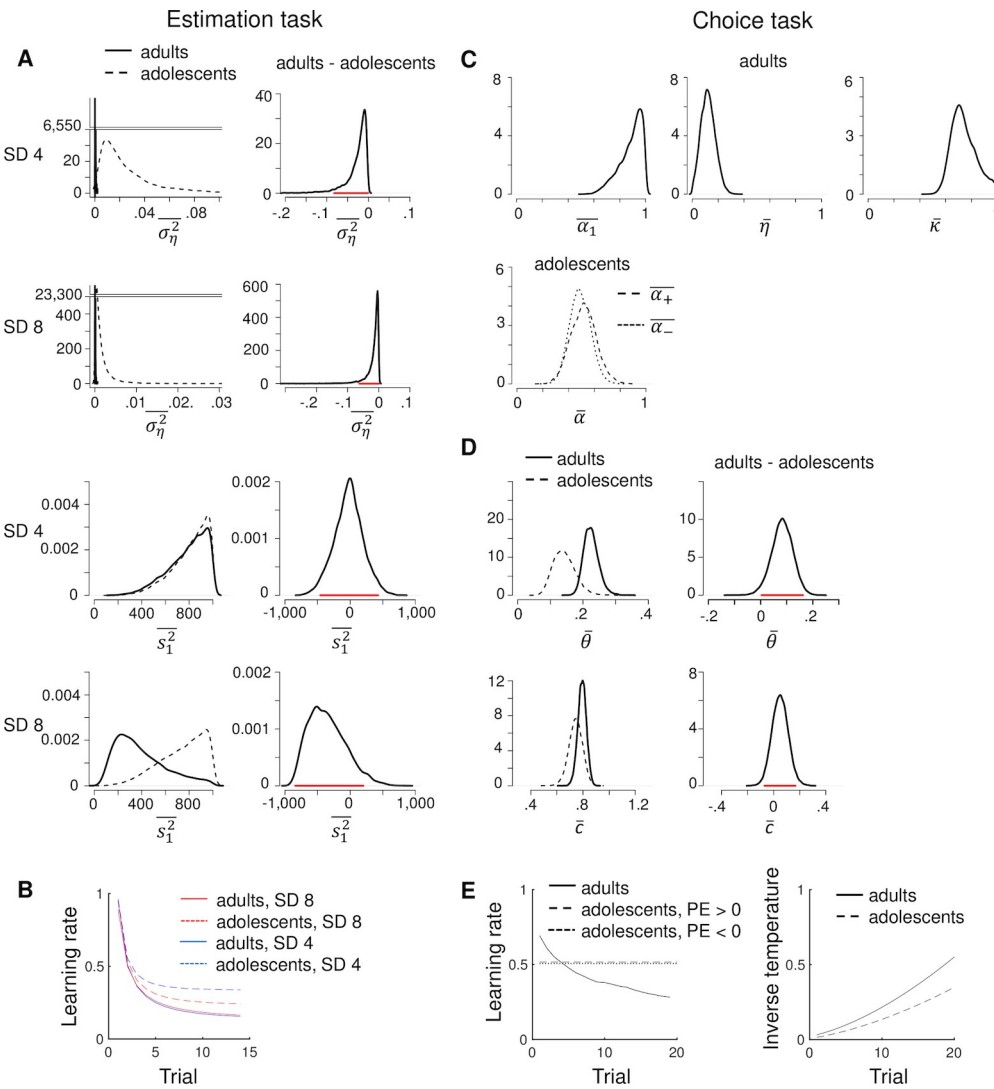

**Fig 5. Parameter estimates. A**. Posterior distributions for the group-level central tendencies of the Kalman-filter parameters per age group (left panels) and the corresponding difference distributions (right panels; red lines are 95% HDIs) for fits to the estimation data. For visualization purposes, we omitted y-axis values between 40 and 6,550 and between 500 and 23,300, respectively, in the two upper left plots. **B**. Model's predicted learning rate per trial for fits to the estimation data. We used the medians of the individual-level posterior distributions to compute trial-specific learning rates per participant, and plotted the average data for each age group. **C**. Posterior distributions for the group-level means of the reinforcement learning/Pearce-Hall hybrid (adults) and asymmetric reinforcement learning (adolescents) parameters for fits to the choice data. The SD of the outcome-generating distribution in the choice task was always 8. **D**. Posterior distributions for the group-level means of the dynamic-softmax parameters per age group (left panels) and the corresponding difference distributions (right panels) for fits to the choice data. **E**. Model's predicted learning rate and inverse temperature per trial for fits to the choice data, computed as in Fig B. PE = prediction error. Note that the adolescents' learning rates following positive and negative prediction errors almost overlap.

those for the adults (Fig 5A, upper left panels). To further examine these group differences, we calculated the difference between the posterior distributions for the adults and adolescents (Fig 5A, right panels). 100 and 99.5% of the resulting difference distribution for the low- and high-noise condition, respectively, lay below 0, suggesting that the adolescents assumed more variability in the mean outcome over time (i.e., higher volatility) than the adults.

The posterior distributions of $\overline{s_1^2}$ (initial prior variance) exceeded 100 for both age groups, suggesting that the first estimate of the mean outcome was extremely uncertain. This makes sense as the first estimate was made before any outcome had been observed, thus could not be based on any prior knowledge (participants basically had to guess on the first trial). Such very high values of $s_1^2$—relative to the noise variance $\sigma_\varepsilon^2$ which was fixed at 1—produce initial learning rates that approach 1 (Eq 9 in the Methods), which is optimal in this task. We did not find evidence that $\overline{s_1^2}$ differed between the two age groups in either noise condition (54 and 88% of the difference distribution for the low- and high-noise condition, respectively, lay below 0).

In the Kalman filter, the ratio between $\sigma_\eta^2$, $s_1^2$ and $\sigma_\varepsilon^2$ determines the level of uncertainty about the mean outcome on each trial ($s_t^2$), which in turn determines the trial-specific learning rate (see Methods). Fig 5B shows the average learning rate timecourse per group and noise level predicted by the model (computed per participant using the medians of the individual-level parameter estimates, and then averaged across participants). Note that the learning rates predicted by the Kalman filter (Fig 5B) show a similar pattern as the learning rates that we directly computed from participants' estimation data (Fig 2B), in terms of differences between the age groups and noise levels (higher asymptotic learning rates for the adolescents than the adults, especially for the low-noise blocks, reflecting the higher drift variance in the adolescents). However, the adolescents' asymptotic learning rates predicted by the model were somewhat lower than those directly computed from the data.

**Choice task.**  Fig 5C shows the posterior distributions of the group-level mean parameters of the best-fitting learning models, when applied to each age group's choice data: the reinforcement learning/Pearce-Hall hybrid model for the adults (upper panel) and the asymmetric reinforcement learning model for the adolescents (lower panel). The adults' choice data was best explained by a model in which learning rate decreased over trials, in a non-Bayesian way, while the adolescents' choice data was best explained by a model with constant learning rates (Fig 5E, left panel). The winning model for the adolescents had separate learning rates for positive and negative prediction errors, but the group-level means of these two learning rates were highly similar (Fig 5C, lower panel; Fig 5E, left panel). This suggests that some adolescents used higher learning rates for positive than negative prediction errors while others showed the opposite bias (at the individual level, $\alpha_+$ was higher than $\alpha_-$ for 16 adolescents, and $\alpha_+$ was lower than $\alpha_-$ for 9 adolescents).

Fig 5D shows the posterior distributions of the group-level mean parameters of the dynamic softmax function, for each age group. Together, parameters $\theta$ and $c$ determine the inverse temperature on each trial (a higher inverse temperature could reflect less exploration, or a better ability of the model to capture the participants' choices). The posterior distribution of $\overline{\theta}$ (inverse temperature halfway a task block) was higher for the adults than the adolescents (Fig 5D; 97.5% of the difference distribution lay above 0), reflecting more exploratory choice behaviour in the adolescents, or a worse ability of the model to capture the adolescents' choices. The posterior distribution of $\overline{c}$, which governs the change in inverse temperature over trials, was positive in both groups, indicating that participants became less exploratory over time, or that the model was better able to capture participants' choices during late than early trials. In addition $\overline{c}$ was numerically higher in the adults than in the adolescents (78.6% of the difference distribution for $\overline{c}$ lay above 0). Fig 5E shows the average inverse temperature timecourse per group predicted by the models. Although the increase in inverse temperature was steeper for the adults than the adolescents, the increase in inverse temperature (inverse temperature on trial 20—inverse temperature on trial 1) did not differ significantly between the two age groups ($t(58) = 1.6$, $p = .12$).

Note that the best-fitting learning models in the choice task differed between the two age groups, which may have affected the group differences in estimated softmax parameters reported above. Therefore, we repeated the group comparisons of $\overline{\theta}$, $\overline{c}$, and inverse temperature, this time using parameter estimates derived from the same model in both age groups: the reinforcement learning/Pearce-Hall hybrid model + dynamic softmax (the winning model for the adults, and shared second best model for the adolescents). This resulted in somewhat stronger group differences for $\overline{\theta}$ and $\overline{c}$: 99.7 and 85.7% of their corresponding difference distributions lay above 0 (S1A Fig). In addition, this analysis revealed that the increase in inverse temperature over trials was significantly larger in the adults than adolescents ($t(58) = 3.1$, $p = .003$; S1B Fig).

## Between-subject relationships between model-based estimates of assumed volatility and exploration

Our model-based analyses of the estimation task suggested that the adolescents assumed a more volatile outcome-generating process than the adults (higher drift variance, $\sigma_\eta^2$, of the Kalman filter), which resulted in higher asymptotic learning rates. In a choice context, higher levels of volatility warrant not only higher learning rates, but also more exploration. Therefore, it is an interesting possibility that the more exploratory choice behaviour in the adolescents was driven by higher assumed volatility as well. Alternatively, it is also possible that the increased exploration in adolescents reflected a different process, not involving volatility assumptions (e.g., reduced sensitivity to value differences, or more error responses). If the age-related differences in exploration were indeed caused by different volatility assumptions, we would expect that (i) participants who assume higher volatility explore more, and (ii) age-related differences in exploration are mediated by individual differences in assumed volatility.

To test these hypotheses, we conduct single-level mediation analyses with age group (adults vs. adolescents) as the predictor, each participant's estimated $\sigma_\eta^2$ parameter from fits to the estimation data (index of assumed volatility) as the mediator, and each participant's estimated $\theta$ and $c$ parameter from fits to the choice data (potential indices of, respectively, the degree of exploration halfway a task block and the decrease in exploration over trials) as the dependent variable, in two separate mediation models (Fig 6A). The individual-level estimates of $\sigma_\eta^2$ contained some strong outliers (most estimates lay below .01, but there were also several participants with estimates above 1, such that $\sigma_\eta^2$ was not normally distributed). To deal with this, we replaced the $\sigma_\eta^2$, $\theta$, and $c$ variables with their ranks.

Consistent with the posterior distributions of $\overline{\sigma_\eta^2}$ reported in the previous section, $\sigma_\eta^2$ was lower in the adults than adolescents (path $a$, $p < .001$, in both mediation models). Importantly, higher values of $\sigma_\eta^2$ (higher assumed volatility in the estimation task) predicted lower values of $\theta$ (more exploration in the choice task, or a worse ability of the model to capture participants' choices) and lower values of $c$ (a smaller decrease in exploration, or a smaller difference between the model's ability to capture early and late choices), when controlling for age group (path $b$, $p < .01$ and $p < .001$, respectively; Fig 6B). Moreover, individual differences in $\sigma_\eta^2$ formally mediated the effects of age group on both $\theta$ and $c$ ($a^*b$, $p < .01$ for both mediation effects). When controlling for $\sigma_\eta^2$, the effect of age group on $\theta$ decreased but was still significant (total effect, $p = .004$; direct effect $p = .03$), implying a partial mediation. The effect of age group on $c$ was marginally significant in the unmediated models (total effect, $p = .08$), but became far from significant when controlling for $\sigma_\eta^2$ (direct effect, $p = .94$). We also repeated these analyses using the estimated $\theta$ and $c$ parameters derived from the same choice-task model for both age groups (the reinforcement learning/Pearce-Hall hybrid model + dynamic

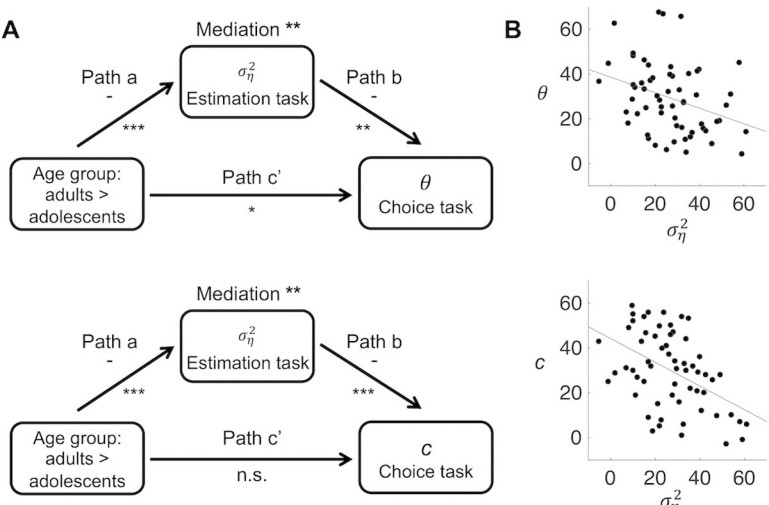

**Fig 6. Relationships between age group and model parameters $\sigma_\eta^2$, $\theta$ and $c$. A**. Mediation models and results. For $\sigma_\eta^2$, $\theta$ and $c$, we used the medians of the posterior distributions of the individual-level parameters, and converted these into ranks. $^{***}$ $p < .001$, $^{**}$ $p < .01$, $^*$ $p < .05$, n.s. not significant. **B**. Across-subject relationships between $\sigma_\eta^2$ and $\theta$, and between $\sigma_\eta^2$ and $c$ (all converted to ranks), controlled for age group (path $b$ of the mediation models). The partial correlations are $r = -.33$, $p = .01$ $r = -.47$, $p < .001$, respectively.

softmax); this yielded the same pattern of significant results (S2 Fig). These findings are consistent with the idea that participants' assumed volatility governed their overall overall degree of exploration as well as their reduction in exploration over time, and that the higher assumed volatility in the adolescents contributed to their increased exploration.

## Validation of modeling results

To validate our model-selection and parameter-estimation results, we performed model-recovery and parameter-recovery analyses, respectively. We describe the main outcomes here; procedures and full results are provided in the Supplementary Material (S3 and S4 Texts and S3–S5 Figs).

**Model recovery.** Model-recovery results indicate that our procedure could distinguish the learning and choice processes implemented in our different models with rather high accuracy, with one exception: a high proportion of choice datasets generated by the reinforcement learning/Pearce-Hall hybrid model + constant softmax was best fit by the standard reinforcement model + constant softmax (discussed in S3 Text and S3 Fig). However, these models were never the best-fitting models in our analysis of the empirical data. Regarding our best-fitting models, the results indicate that (i) we can be 94% confident about the best-fitting model for the estimation task (Kalman filter, in both age groups); (ii) we can be 89% confident about the best-fitting model for the choice task in the adolescents (asymmetric reinforcement learning model + dynamic softmax); (iii) we can be 71% confident about the best-fitting model for the choice task in the adults (reinforcement learning/Pearce-Hall hybrid model + dynamic softmax); and (iv) we can be 90–97% confident about the best fitting softmax function in both age groups (dynamic softmax).

**Parameter recovery.** For the estimation-task models, our parameter-recovery results corroborate the presence of an age-related difference in the Kalman filter's drift-variance parameter $\sigma_\eta^2$. However, while values of $\overline{\sigma_\eta^2}$ corresponding to the estimated value for the adult group were accurately recovered, values of $\overline{\sigma_\eta^2}$ corresponding to the estimated value for the adolescent

group tended to be underestimated (S4 Fig). Thus, the actual value of $\overline{\sigma_\eta^2}$ for the adolescent group may well have been higher—and hence the age-related difference in $\overline{\sigma_\eta^2}$ larger—than it seemed based on the parameter estimates from our model fits to the empirical data.

Regarding the choice-task models, all group-level mean parameters of the asymmetric reinforcement learning model + dynamic softmax—the best-fitting model for the adolescents—were well recovered (correlations between simulated and recovered values > .92, $p$'s < .001, for all parameters; S5 Fig), with no evidence for bias or tradeoffs between parameters. For the reinforcement learning/Pearce-Hall hybrid model + dynamic softmax—the best fitting model for the adults—simulated and recovered parameters were correlated as well (correlations range from .4 to .94, all $p$'s < .005). However, $\overline{\eta}$, $\overline{\kappa}$ and $\overline{c}$ tended to be underestimated, whereas $\overline{\theta}$ tended to be overestimated for this model (S5B Fig). Thus, the estimated values of these hyperparameters should be taken with some caution.

## Discussion

Although uncertainty is an important mediator of learning and exploration [15, 17, 18], previous developmental studies on experience-based learning have not incorporated trial-to-trial changes in uncertainty in their analyses and models. In the present study, we examined uncertainty-driven regulation of learning rate and exploration in adolescents and adults, which provided novel (computational) insights into the nature of developmental changes in experience-based learning.

We found that adolescents, as compared to adults, were characterized by higher asymptotic learning rates in an estimation task and by more exploration in a choice task. Computational modeling provided evidence that both of these effects arose because the adolescents assumed more variation in the outcome-generating process over time, and thus were less certain about their estimates. However, self-reported certainty during the estimation task, and its trial-to-trial relationship with learning rate, did not differ between adolescents and adults.

Taken together, our results suggest that adolescents, as compared to adults, overestimate environmental volatility (unpredictable change in stimulus-outcome or action-outcome associations), making them more sensitive to recent relative to older evidence. In Bayesian learning models, this corresponds to weaker priors—less precise belief distributions—resulting in faster belief updating based on new information [34, 35]. As an increased perception of volatility might lead adolescents to consider past experiences as less representative of the future, and experience action-outcome associations as less stable, it may well contribute to the surge in real-life exploration and risk-taking behaviours that are characteristic of adolescence [36–38]. It may also underlie adolescents' stronger tendency to shift between choice options regardless of their expected values, which has been observed in some previous probabilistic learning studies that compared adolescents and adults [9, 12]. As the period of adolescence is characterized by major physical and social change, and rapidly changing environmental demands, an increased perception of volatility during this period may be functionally adaptive by promoting cognitive and behavioural flexibility. Previous work has suggested similar advantages of the increase in some types of risk-taking and sensation-seeking behaviours during adolescence [39–41]. Our results contribute to this discussion by suggesting a computational principle—an increased representation of volatility—that may underlie these behaviours.

Volatility is one form of uncertainty that has been dissociated—both computationally and in terms of relevant brain systems—from noise (stochasticity inherent in the environment) and estimation uncertainty (the imprecision of the learner's current estimate of the environment, due to limited knowledge, which decreases with sampling) [14, 16, 42]. Importantly, learners' estimates of the different forms of uncertainty are not fully independent from each

other. Specifically, when learners assume a higher noise level, they will be more likely to attribute prediction errors to random outcome fluctuations rather than to a change in outcome contingencies, resulting in lower perceived volatility. In that sense, the evaluations of volatility and noise can be considered antagonistic [16], which is also why their ratio—and not their absolute values—matters in the Kalman filter. Thus, an underestimation of noise in the adolescents—which made them more inclined to interpret unexpected outcomes as indicative of change—may have contributed to their overestimation of volatility. Future studies are required to further disentangle the roles of these different forms of uncertainty in learning across development, for example by independently manipulating volatility and noise.

Our finding that self-reported certainty during the estimation task did not differ between the adolescents and adults seems at odds with the higher learning rates in the adolescents, as normative learning implies that higher learning rates should be associated with higher uncertainty. What might explain this apparent discrepancy? First, this may be explained by the interdependence of volatility and noise described above: If adolescents assume more volatility than adults, but also less noise, this could result in increased learning rates (which scales with the ratio between volatility and noise) despite the same 'overall' certainty experience (which scales with their sum). Second, the evaluation of the various forms of uncertainty is likely to be an (at least partially) implicit process. Indeed, our model-based analyses revealed that participants (especially the adolescents) assumed variability in outcome contingencies over time even though they were instructed that this was not the case, suggesting a dissociation between participants' explicit task knowledge and the internal model they used to perform the task. Third, it has been proposed that self-evaluation of performance involves a distinct decision system that "reads out" confidence information from internal states, and that substantial bias can be introduced during this process [43, 44]. Thus, age-related differences in this reporting bias may also explain why adolescents reported to be as certain as adults while their learning rates suggested otherwise. Finally, that the adolescents reported to be equally certain as the adults, even though their behaviour suggested that they judged the environment to be more volatile, may indicate that the adolescents were actually more confident about their performance than the adults. This last possibility is broadly consistent with some previous developmental studies on performance monitoring [45, 46].

By directly comparing an estimation and a choice task, our study also provides evidence that estimating the mean value of a stimulus engages a different learning process than making choices between two stimuli. The estimation data of both age groups was best described by a learning model that integrates new outcomes and prior expectations using Bayesian inference (Kalman filter). In contrast, the choice data was best described by simpler, non-Bayesian, learning processes (reinforcement learning/Pearce-Hall hybrid model for the adults, and asymmetric reinforcement learning model for the adolescents). Thus, during the choice task, we found evidence that adults decreased their learning rate over time—although in a different way than in the estimation task—whereas adolescents used constant learning rates. Possible reasons for the use of simpler, non-Bayesian, learning processes in the choice task are that (i) optimal performance in this task requires estimating the *relative* rather than the *exact* value of stimuli and/or (ii) optimal choice performance can be achieved by a simpler updating process when combined with a dynamic adjustment of the exploration-exploitation tradeoff. In addition, the choice task involved learning about two stimuli in parallel (compared to one stimulus in the estimation task), requiring more working-memory resources, which may have resulted in the use of simpler learning processes. Indeed, estimation performance on a more computationally demanding estimation task in a previous study—adult participants learned about two stimuli during each task block—was best explained by a non-Bayesian learning model that resembled our best-fitting model for the adults' choice data [47].

Because the outcome-generating process in our experimental tasks was static (i.e., actual volatility was zero), adolescents' elevated learning and exploration rates were sub-optimal in these tasks, deteriorating performance. In certain types of dynamic environments, however, these same learning characteristics may be advantageous by promoting the detection of, and adaptation to, change. Thus, an important objective for future studies is to examine whether and how the increased representation of volatility in adolescents identified in the present study affects their learning performance in changing environments. This is presumably dependent on the type of environmental change. When outcome contingencies change in a continuous manner—e.g., according to a diffusion process—an increased representation of volatility might be beneficial, especially when the rate of change is fast. In contrast, in environments containing discrete 'change points', an increased representation of volatility may be disadvantageous as it could lead to the over-estimation of change-point frequency, and excessive expectation updating during stable periods in between change points. These ideas remain to be tested in future studies that compare adolescents' and adults' performance on learning tasks involving continuous [48, 49] versus discrete [50, 51] change.

Another open question concerns the use of uncertainty in learning and exploration prior to adolescence. As our study focused on adolescents and adults, it leaves open the question whether the overestimation of volatility is specific to the period of adolescence, or also present—and perhaps even stronger—in younger children. It has been shown that young children are in some cases more flexible learners—more willing to adopt hypotheses that are supported by new evidence but inconsistent with preexisting societal notions—than older children and adults, possibly because they are less biased by prior knowledge and beliefs [52–54]. Interestingly, one recent study that included children, adolescents and adults found that the ability to learn unusual physical causal relationships decreased monotonically with age, while learning of unusual social causal relationships peaked at the age of four and again during adolescence [55]. Although the latter study did not manipulate or measure uncertainty, and focused on the outcome of learning rather than the learning process itself, its results suggest that the sensitivity to new evidence is highest in both preschoolers and—at least in the social domain—adolescents. Whether or not the same underlying process—e.g., an increased volatility assumption—mediated this effect in both of these age groups remains to be explored in future work.

Finally, a recent study in adults showed that, in a restless two-armed bandit task, a substantial proportion of choices for the lower-valued option can be explained by internal computational noise in the expectation-updating process—resulting in random fluctuations in expected outcome values—rather than by exploratory choice behaviour [56]. An interesting idea that follows—and remains to be tested—is that not only learning rate and exploration, but also 'learning noise', increase as a function of environmental volatility. If this is the case, adolescents' increased representation of volatility could have resulted in a noisier learning process in this age group. Whether, and to what extent, age-related differences in learning noise may underlie the age-related effects on learning rate and exploration found in the present study needs to be examined in future work, for example using the recently developed 'noisy RL model' [56].

To conclude, our findings suggest that adolescents assume a faster-changing world than adults, resulting in more malleable expectations and more persistent exploration. Future studies may examine the potentially adaptive and maladaptive consequences of these learning features in different types of environments, which may eventually inform educational strategies and policy interventions aimed at adolescents.

## Methods

### Ethics statement

All procedures were approved by the local ethics committee of the Faculty of Social and Behavioural Sciences of the University of Amsterdam. Adult participants provided written informed consent. For the adolescents, primary caretakers were informed about the experiment and provided written informed consent for the participation of their child.

### Participants

Thirty-five young adults (mean age = 22.6, range = 18–29; 77% female) and twenty-five adolescents (mean age = 13.7, range = 12–15; 64% female) participated in the study. Adults were students, or former students, at universities or colleges of higher professional education. Adolescents attended high schools (pre-university or higher general secondary education). All participants were healthy and reported no history of psychiatric or neurological disorders, and no use of alcohol or recreational drugs on the test day. Participants received €8 or (some of the adult participants) course credits for their participation. In addition, all participants earned a performance-related bonus of 0 to 2 euros per task.

### General procedure

Participants individually completed the estimation and the choice task, respectively, which are described below. Before starting each task, participants were informed about the task structure and procedure by means of computerized instructions. The estimation and choice tasks lasted approximately 30 and 15 minutes, respectively.

### Estimation task

During this task, a box was presented on the screen and participants repeatedly observed a number outcome—corresponding to a number of points—inside the box. The number on each trial was randomly drawn from a Gaussian distribution with a specific mean and standard deviation. The mean of the number-generating Gaussian distribution was fixed within each experimental block, but varied across blocks (between 20 and 80; Fig 1A). The standard deviation (SD) of the distribution was 4 (low noise) in the first half of the blocks, and 8 (high noise) in the second half of the blocks, or vice versa, in counterbalanced order. The start of each new block was clearly indicated, by means of a 'new-block' screen, and a different box, with a unique color, was used in each block.

Participants completed ten experimental blocks, of fifteen trials each. At the start of each trial, participants estimated the average number of points inside the current box as closely as possible, by typing a number on the keyboard (self-paced; Fig 1B). The typed number appeared underneath the box, and participants confirmed their estimate by pressing the ENTER key. Participants then rated their certainty about their estimate on a vertical scale from 0 to 10 with lower and upper anchors of "completely uncertain" and "completely certain", respectively, using the mouse (self-paced). Then a number was randomly drawn from the current Gaussian distribution and displayed inside the box for 1.5 seconds. After that, the box closed again and 1 second later the next trial started.

Except for the specific means and SDs used, participants were fully informed about the task structure and procedure before starting the task. Specifically, to provide an intuitive explanation that the numbers were generated from a stable but noisy process, we instructed participants that the number of points inside a box would vary from time to time, but that the average number of points would not change during a block. Participants were also instructed

that all numbers would fall within the range of 1 through 100, and that the average number of points differed between blocks. After block 5, participants were instructed that the variability in numbers would increase or decrease from then on, corresponding to the change in the SD of the number-generating distribution. In addition, blocks 1 and 6 were preceded by a 15-trial practice block with the same SD as the upcoming experimental blocks, to familiarize participants with the outcome variability in the upcoming block. Finally, participants were instructed that they could earn a bonus of maximally 2 euros depending on how accurate their estimates were.

**Computation of trial-specific learning rate.** We estimated the prediction error ($\hat{\delta}$) on each trial $t$ as the difference between the actual outcome and the participant's estimate on that trial: $\hat{\delta}_t = Outcome_t - Estimate_t$. Consequently, we estimated trial-specific learning rate ($\hat{\alpha}$) as the change in the participant's estimate across two successive trials, trial $t$ and $t+1$, divided by the prediction error on trial $t$: $\hat{\alpha}_t = (Estimate_{t+1} - Estimate_t)/\hat{\delta}_t$ [19, 20].

## Choice task

During this task, participants made repeated choices between two options (boxes), presented at the left and right side of the screen (Fig 1C). Each time a box was chosen, it paid off a number of points, and participants' task was to earn as many points as possible. As in the Estimation task, the number of points corresponding to each box was randomly drawn from a specific Gaussian distribution. Participants completed eight 20-trial blocks of this task. The difference in the means of the Gaussian distributions associated with the two boxes was either ten or twenty points, in half of the experimental blocks each. Specifically, the distribution associated with one of the boxes was centered at 40, 50 or 60, and the distribution associated with the other box was centered either ten or twenty point higher or lower, in random order. The SD of all distributions was 8.

At the start of each trial, participants selected the left or the right box, by pressing a left or a right key. Then a number was randomly drawn from the Gaussian distribution corresponding to the chosen box and displayed inside that box for 1.5 seconds. After that, the box closed again, and 1 second later the next trial started.

As in the Estimation task, participants were fully informed about the task structure and procedure—except for the specific means and SDs used—and completed a 20-trial practice block, before starting the first experimental block. In addition, participants were instructed that they could earn a bonus of maximally 2 euros depending on how often they chose the box that paid off most points on average.

## Behavioural analysis

We performed multilevel linear regression and mediation analyses on single-trial estimates of learning rate ($\hat{\alpha}$) and certainty ratings during the Estimation task, and multilevel logistic regression on single-trial choice accuracy during the Choice task. Random intercepts and slopes were modeled in all analyses, and covariances between random effects were fixed to 0. In the linear regression models, correlation between error terms across trials was considered by specifying first-order autoregression. Linear and logistic regression analyses were performed in R, using the nlme [57] and lme4 [58] packages, respectively. Mediation analysis was performed in matlab, using the Multilevel Mediation toolbox (http://wagerlab.colorado.edu/tools [59]).

**Analysis of the estimation task.** To examine changes in learning rate and certainty as a function of the number of observed outcomes, we tested for the linear and quadratic effects of

block-specific trial on learning rate and certainty ratings, in two separate regression models. In both analyses, we also modeled the SD of the number-generating distribution (high vs. low noise) and age group (adults vs. adolescents; second-level regressor), and all trial x SD x age group interactions.

To further examine the relationships between the number of observed outcomes, certainty and learning rate, we performed a multilevel mediation analysis. In our mediation model, we used block-specific trial as the independent variable, learning rate as the dependent variable, and certainty rating as the mediator, and included age group as second-level moderator. We used bootstrapping (100,000 bootstrap samples) for significance testing.

In the analyses on learning rate, we excluded the last trial of each block, and trials on which the prediction error ($\hat{\delta}$) was equal to 0 (5.5% and 4.5% of all trials in the adult and adolescent group, respectively), as learning rate could not be computed on those trials. In the regression analysis on certainty, we excluded the first certainty rating, which was made before any outcomes were observed. In addition, we excluded trials with estimates of 0 or above 100 from all analyses (0.3% and 0.9% of all trials in the adult and adolescent group, respectively), as these presumably reflected typing errors (as participants were instructed that all numbers would be in the range of 1 to 100). Finally, inspection of the data suggested that some estimates below 100 also reflected typing errors (e.g., when a participant accidentally omitted the first digit and typed '3' instead of '83'), which could produce extreme learning-rate estimates. To deal with this, we excluded trials on which the estimated learning rate exceeded the 99[th] percentile or was lower than the 1[st] percentile (calculated separately for the adult and adolescent group; 1.7% and 1.8% of all learning rates in the two groups, respectively).

**Analysis of the choice task.**   We tested for the linear and quadratic effects of block-specific trial on choice accuracy, using multilevel logistic regression. We also modeled the effect of age group and the trial x age group interactions.

## Computational models

**Standard reinforcement learning model.**   This model updates expectations about stimulus-outcome associations in response to each new observed outcome, in proportion to the prediction error. On each trial, $t$, the discrepancy between the outcome for the current stimulus $s$, $O_{s,t}$, and the expected outcome at the onset of that trial, $E_{s,t}$, elicits a prediction error, $\delta_{s,t}$:

$$\delta_{s,t} = O_{s,t} - E_{s,t} \tag{1}$$

This prediction error triggers expectation updating according to a standard reinforcement-learning algorithm ('delta rule' learning):

$$E_{s,t+1} = E_{s,t} + \alpha\delta_{s,t} \tag{2}$$

Note that, given Eq 1, this updating process can also be written as:

$$E_{s,t+1} = \alpha O_{s,t} + (1 - \alpha)E_{s,t} \tag{3}$$

which shows that the new expectation is a weighted average of the most recent observation and the old expectation. Learning-rate parameter $\alpha$ determines their relative weights—i.e., the amount of updating—such that higher values of $\alpha$ result in stronger updating.

**Asymmetric reinforcement-learning model.**   This model is identical to the standard reinforcement learning model, with the following exception: it uses two separate learning rates ($\alpha_+$ and $\alpha_-$) to update expectations following positive and negative prediction errors, hence Eq 2 is

replaced by:

$$E_{s,t+1} = \begin{cases} E_{s,t} + \alpha_+ \delta_{s,t} \; \text{if} \; \delta_{s,t} > 0 \\ E_{s,t} + \alpha_- \delta_{s,t} \; \text{if} \; \delta_{s,t} < 0 \end{cases} \tag{4}$$

**Kalman filter.**   This model represents learning as Bayesian inference. As a Bayesian model, the Kalman filter attributes to the participant a structured belief—i.e., a generative model—regarding the statistics and dynamics of the task environment. The participant is assumed to engage in optimal Bayesian inference with respect to this generative model, based on the observed outcomes.

According to the generative model, the outcome on each trial, $t$, for stimulus $s$, $O_{s,t}$, is sampled from a Gaussian distribution with a fixed SD, $\sigma_\varepsilon$ ($\sigma_\varepsilon$ represents outcome noise). The generative model also assumes that the mean of the Gaussian distribution associated with each stimulus, $x_{s,t}$ varies over time according to a Gaussian random walk with mean step size 0 and SD $\sigma_\eta$, adding some uncertainty to the estimate of the mean after each trial. The actual outcome-generating distributions in our tasks did not change over time, but it is possible that participants (implicitly) assumed some variation of the mean in their generative model, which could account for suboptimal learning-rate adaptation. Specifically, when $\sigma_\eta$ is larger than 0, this causes over-weighting of the most recent outcome, hence elevated learning rates.

We assume participants maintain a conjugate iterative prior on $x_{s,t}$ conditioned on all previous observations for stimulus $s$:

$$x_{s,t} | O_{s,t-1} \sim \mathcal{N}(\mu_{s,t}, s_{s,t}^2) \tag{5}$$

where $O_{s,t-1} = \{O_{s,1}, \ldots, O_{s,t-1}\}$ is the sequence of all outcomes observed for stimulus $s$ so far. The mean and variance of this prior—$\mu_s$ and $s_s^2$, respectively—are values tracked by the participant: $\mu_s$ is the participant's estimate of $x_s$, and $s_s^2$ is the participant's uncertainty about that estimate. Following each new observed outcome, a posterior belief about $x_{s,t}$ is calculated. The mean of this posterior is a precision-weighted average of the prior estimate and the new observation:

$$x_{s,t} | O_{s,t} \sim \mathcal{N}\left( \frac{\sigma_\varepsilon^2 \mu_{s,t} + s_{s,t}^2 O_{s,t}}{\sigma_\varepsilon^2 + s_{s,t}^2}, \frac{\sigma_\varepsilon^2 s_{s,t}^2}{\sigma_\varepsilon^2 + s_{s,t}^2} \right) \tag{6}$$

The prior on the next trial is then obtained by adding the variance of the random walk, $\sigma_\eta^2$:

$$x_{s,t+1} | O_{s,t} \sim \mathcal{N}\left( \frac{\sigma_\varepsilon^2 \mu_{s,t} + s_{s,t}^2 O_{s,t}}{\sigma_\varepsilon^2 + s_{s,t}^2}, \frac{\sigma_\varepsilon^2 s_{s,t}^2}{\sigma_\varepsilon^2 + s_{s,t}^2} + \sigma_\eta^2 \right) \tag{7}$$

By definition, the mean and variance of this iterative prior equal $\mu_{s,t+1}$ and $s_{s,t+1}^2$, respectively. This relation yields the update equation for $\mu$:

$$\mu_{s,t+1} = \frac{\sigma_\varepsilon^2}{\sigma_\varepsilon^2 + s_{s,t}^2} \mu_{s,t} + \frac{s_{s,t}^2}{\sigma_\varepsilon^2 + s_{s,t}^2} O_{s,t} \tag{8}$$

Thus, as in the reinforcement learning model, the new expectation is a weighted average of the most recent outcome and the old expectation. To see the connection formally, define a

trial-specific learning rate:

$$\alpha_{s,t} = \frac{s_{s,t}^2}{\sigma_{\varepsilon}^2 + s_{s,t}^2} \tag{9}$$

Then the update equation for $\mu$ can be written as:

$$\mu_{s,t+1} = \alpha_{s,t} O_{s,t} + \left(1 - \alpha_{s,t}\right)\mu_{s,t} \tag{10}$$

in agreement with the expectation updating rule in the reinforcement learning model. Thus, both models use the same learning principle, but the Kalman filter determines the learning rate rationally and dynamically, based on the assumed variance of the outcome-generating distribution ($\sigma_{\varepsilon}^2$) and the current level of uncertainty in the prior for $x_{s,t}$, or estimation uncertainty ($s_{s,t}^2$).

The Kalman filter has 3 free parameters: $\sigma_{\varepsilon}^2$ (noise variance), $\sigma_{\eta}^2$ (drift variance), and $s_{s,1}^2$ (the initial variance of the prior for $x$). Only the ratios between these three parameters matter; hence we fixed $\sigma_{\varepsilon}^2$ at a value of 1 and estimated $\sigma_{\eta}^2$ and $s_{s,1}^2$.

**Reinforcement learning/Pearce-Hall hybrid model.**   This model scales the learning rate by a dynamic 'associability' term which originates from the Pearce-Hall model [28]. The associability on trial $t$, $\alpha_t$, is a weighted average of the absolute prediction error and associability on the previous trial:

$$\alpha_{s,t} = \eta|\delta_{s,t-1}| + (1 - \eta)\alpha_{s,t-1} \tag{11}$$

Decay parameter $\eta$ determines the relative weights of these two terms. The associability term is incorporated in the expectation updating algorithm from the standard reinforcement learning model [29–31]:

$$E_{s,t+1} = E_{s,t} + \kappa\alpha_{s,t}\delta_{s,t} \tag{12}$$

The initial associability (on trial 1; $\alpha_1$) is estimated as a free parameter. Thus, this model has three free parameters: $\alpha_1$, $\eta$, and $\kappa$.

For fits to the estimation data, parameters from all models were estimated separately for the low and high noise conditions, and we initialized expectations ($E_{s,1}$ in models 1, 2 and 4 and $\mu_{s,1}$ in model 3) to participants' first prediction (separately for each particiant and block). For fits to the choice data, we initialized all expectations to 50 (the center of the range of possible outcomes).

**Decision function.**   For fits to the choice data, we combined both learning models with a softmax function, which computes the probability of choosing stimulus $s$ on trial $t$ ($P_{s,t}$) as:

$$P_{s,t} = \frac{e^{E_{s,t}*\beta}}{\sum_{s'=1}^{2} e^{E_{s',t}*\beta}} \tag{13}$$

Inverse-temperature parameter $\beta$ controls the sensitivity of choice probabilities to the differences in expected value ($E_{s,t}$ and $\mu_{s,t}$ in the reinforcement learning and Kalman filter model, respectively) between the two options. If $\beta$ is 0, both options are equally likely to be chosen, irrespective of their expected values (reflecting a high degree of choice randomness or exploration, or an inability of the model to capture participants' choices). As the value of $\beta$ increases, the probability that the stimulus with the highest expected value will be chosen increases (higher degree of exploitation, or better ability of the model to capture participants' choices).

We compared two versions of the softmax function. In the first version (constant softmax; Model 1A and 2A), $\beta$ was constant over trials. In the second version (dynamic softmax; Model

1B and 2B), $\beta$ changes as a function of the number of previous trials, according to:

$$\beta_t = \theta * (t/10)^c \tag{14}$$

Response-consistency parameter $c$ controls the change in $\beta$ over time: Positive and negative values of $c$ result in an increase and decrease in $\beta$ over time, respectively, and larger absolute values of $c$ result in faster changes. Finally, parameter $\theta$ determines the value of $\beta$ on the 10th trial.

### Parameter estimation

We applied the models to participants' trial-to-trial sequences of estimates and choices—for the estimation and choice task, respectively—using a hierarchical Bayesian approach. Before fitting the model to the estimation data, we replaced estimates of 0 and estimates above 100 (which likely reflected typing errors) with the average of the immediately preceding and following estimate. The hierarchical procedure assumes every participant has a different set of model parameters, drawn from a group-level distribution [60]. In a Bayesian framework, this means that each individual-level parameter is assigned a group-level prior distribution, whose distribution parameters (hyperparameters) are also assigned prior distributions (hyperpriors). We estimated separate group-level distributions for the adult and adolescent group. As we are primarily interested in average-participant behaviour within each age group, and in potential group differences, our primary variables of interest are the hyperparameters governing the central tendencies of the group-level distributions (i.e., the mean and scale parameters governing, respectively, the beta and half-Cauchy group-level distributions).

**Prior distributions.**   Group-level distributions for all model parameters, except the Kalman-filter paramers, were assumed to be beta distributions. Beta distributions are typically defined by two shape parameters ($a$ and $b$), but we reparameterized these in terms of a group-level mean ($a/[a+b]$) and group-level precision ($a+b$) [61]. The group-level means were assigned a uniform hyperprior on the interval [0,1] and the logarithms of the group-level precisions were assigned a uniform hyperprior on the interval [log(2), log(600)] [62]. We transformed the range of the individual-level $c$ parameter (which governs the change in inverse temperature over trials) from [0, 1], the range of the beta distribution, to [-2, 2] using the following transformation: $c = c^* * 4 - 2$ [63]. Group-level distributions for Kalman-filter parameters $\sigma_\eta^2$ and $s_{s,1}^2$ (drift variance and initial prior variance) were assumed to be half-Cauchy distributions. For fits to the estimation data, participants' estimates were assumed to be normally distributed around the models' point predictions. The variance of this normal distributions (error variance) was assumed to have a half-Cauchy group-level distribution. The scale parameters governing the half-Cauchy distributions were assigned a uniform hyperprior on the interval [0,1000]. These priors and hyperpriors were chosen for their uninformative nature [64].

**Parameter-estimation details.**   We inferred posterior distributions for all model parameters using Markov chain Monte Carlo (MCMC) sampling, as implemented in the JAGS package [65] via the R2jags interface. We ran 3 independent MCMC chains with different starting values per model parameter, and collected 40,000 posterior samples per chain. We discarded the first 20,000 iterations of each chain as burn-in. In addition, we only used every 5th iteration to remove autocorrelation. Consequently, we obtained 12,000 representative samples per parameter per model. When using this procedure for fits to the choice data, we encountered convergence problems (R-hat value > 1.1) for a few individual-level parameters of Model 2b (Kalman filter + dynamic softmax). Therefore, we collected twice as many posterior samples for this model (24,000 after burn-in and thinning), after which the chains showed convergence.

**Model comparison.** We compared the fit of the different models using the deviance information criterion (DIC) [32], separately for the two age groups and tasks. The DIC provides an index of the goodness of fit of a model, penalized by its effective number of parameters. Models with smaller DIC are better supported by the data.

## Supporting information

**S1 Text. Deviations from preregistration.**
(DOCX)

**S2 Text. Optimal adaptation of learning rate in noisy but static environments.**
(DOCX)

**S3 Text. Model recovery analysis.**
(DOCX)

**S4 Text. Parameter recovery analysis.**
(DOCX)

**S1 Fig. Control analyses on the dynamic-softmax parameters and inverse temperature.**
(DOCX)

**S2 Fig. Control analyses on the relationships between age group and parameters $\sigma_\eta^2$, $\theta$ and $c$.**
(DOCX)

**S3 Fig. Model-recovery results.**
(DOCX)

**S4 Fig. Parameter-recovery results for the estimation task.**
(DOCX)

**S5 Fig. Parameter-recovery results for the choice task.**
(DOCX)

**S1 Table. The average probability of participants' choices (standard deviations in parentheses) under each model.**
(DOCX)

**S2 Table. Medians (and 95% highest density intervals) of the posterior distributions shown in Fig 5A in the main text.**
(DOCX)

**S3 Table. Prior distributions for the simulation hyperparameters used in the model-recovery analysis.**
(DOCX)

## Acknowledgments

We thank Maarten Speekenbrink for helpful discussions.

## Author Contributions

**Conceptualization:** Marieke Jepma, Jessica V. Schaaf, Ingmar Visser, Hilde M. Huizenga.

**Data curation:** Marieke Jepma, Jessica V. Schaaf.

**Formal analysis:** Marieke Jepma.

**Funding acquisition:** Hilde M. Huizenga.

**Investigation:** Marieke Jepma, Jessica V. Schaaf.

**Methodology:** Marieke Jepma.

**Project administration:** Marieke Jepma.

**Supervision:** Hilde M. Huizenga.

**Visualization:** Marieke Jepma.

**Writing – original draft:** Marieke Jepma.

**Writing – review & editing:** Jessica V. Schaaf, Ingmar Visser, Hilde M. Huizenga.

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
