## [Decision Letter · Decision Letter 0]

17 Dec 2019

Dear Dr Jepma,

Thank you very much for submitting your manuscript 'Uncertainty-driven regulation of learning and exploration in adolescents: A computational account' for review by PLOS Computational Biology. Your manuscript has been fully evaluated by the PLOS Computational Biology editorial team and in this case also by two independent peer reviewers. The reviewers appreciated the attention to an important question, but raised some substantial concerns about the manuscript as it currently stands. While your manuscript cannot be accepted in its present form, we are willing to consider a revised version in which the issues raised by the reviewers have been adequately addressed. We cannot, of course, promise publication at that time.

*** You will see that both reviewers make similar comments. In particular, they are challenging your model-based analysis. More precisely, they are highlighting the fact that your interpretation of the observed difference in learning rate between the two groups may be overly restrictive. Recall that the Kalman filter can only explain this difference in terms of perceived volatility of the environment (under the assumption that model recovery is accurate). In other terms, for a man with a hammer, everything looks like a nail (figuratively speaking). Now you should explicitly look for alternative explanations of this difference. Practically speaking, this means performing (more) model comparisons, having included in the comparison set models that can a priori explain this difference. Note: you should also perform confusion analyses (to assess the accuracy of model comparisons) and parameter recovery analyses (at least for the winning model). I hope you will find these comments helpful. *** 

Sincerely,

Jean Daunizeau

Associate Editor

PLOS Computational Biology

Natalia Komarova

Deputy Editor

PLOS Computational Biology

[LINK]

Reviewer's Responses to Questions

**Comments to the Authors:**

Reviewer #1: Review for “uncertainty-driven regulation of learning and exploration in adolescence” by Jepma et al. The authors performed an experiment involving adults (N=35) and adolescents (N=25) and two learning tasks: a value estimation task and two-armed bandit. The authors report that the adolescents underperform in both tasks by displaying: i) exaggerate learning rate and ii) exploration. They propose that the behaviour in these is well accounted by a Kalman filter model (instead of a standard RL model) and that their results indicate that adolescents over-estimate the volatility of the environment. I think the paper has the potential to deliver an important message to the cognitive developmental community, however, I believe that several issues have to be addressed in order to support their claims.

1/ the first point concerns the model space. The authors limit model comparison to a standard Rescorla-Wagner model and the Kalman-filter model. Learning in this kind of situations have been studied extensively and a lot of (psychologically sound) models have been proposed and validated. They should be included in the model space. These models include:

An asymmetric reinforcement learning model (see Lefebvre et al.2017 and Sharot’s work) for we which we have good reasons to think that there will be differences between adults and adolescents (see Van Den Bos, Cerebral Cortex, 2013). My prediction is that the noise variance parameter will significantly correlate with the difference in learning rates in this model.

A Pearce-Hall model (which is a simpler, non Baysian, instantiation of the idea that the learning rate is dynamically modulated).

Another important model (even if harder to fit) is the noisy-RL model recently proposed by Findling et al. (nature neuroscience), as this noisy update could explain softmax modulations.

2/ the second point is that the model-comparison results (concerning the old and the new models) should be backed-up by a model recovery analysis, where the authors show that, in simulated datasets, their are capable to retrieve the correct, generative, model. The procedure should be extended to the parameters of the best fitting model to ensure that the conclusions based on parameters comparison are justified.

3/ it is unclear which aspect of the behaviour (especially in the bandit task) IS NOT captured by the standard model. The only model-free results that are displayed are the learning curves (figure 4) and this difference can be perfectly captured by a standard model. Is there more information in the behaviour that can help us discriminate the models?

4/ it would be important to know what is the correlation between the parameters estimated in the estimation and in the learning task. Is there some common ground? Are the two tasks tapping into the same processes? I suspect that they don’t and that different models (and different biases) apply to the two tasks.

5/ it is important to provide the instructions given to the subjects, as it is important to know whether or not the subjects were primed about the frequent changes in contingencies. This would change the interpretation of the results, as it could simply be that adolescents do not follow instructions.

6/ what is the rational to fix the noise variance parameter? How does its inclusion change the results (I guess it favour the Kalman filter because it reduces its number of free parameters)?

7/ how have the authors verified that the two groups were matched in terms of IQ and socio-economic status?

Reviewer #2: Jepma and colleagues studied adolescents and adults on an estimation task and a 2-armed bandit task, varying the noise (standard deviation) of outcomes across blocks, and assessing learning rate and confidence on a single trial basis. They show that while evaluation of uncertainty (confidence) and its effects on decreasing learning rate do not differ between the age groups, younger participants asymptoted to a higher learning rate and showed lower choice accuracy. Model fitting suggests that these behavioral findings may be attributed to adolescents' belief that the environment is constantly changing, and therefore new information is more relevant to future prediction and choice than past knowledge.

I found this paper very well written, with clear exposition of the computational terms and models, and a compelling set of results. I was especially impressed by the explanations of the Kalman Filter, which are the clearest I have encountered. The experiments are clean and well chosen, and the results described in detail, with well-thought-out statistical analyses.

My main worry regards the discrepancy between the behavioral results and the conclusions from the modeling: while the behavioral results suggested no age-related differences in uncertainty and how it changed over time, and in the process of updating learning rates with experience, the results of the modeling were considered as suggesting the opposite. That is, based on the modeling results, it was suggested that adolescents assume more variation in the outcome-generating process over time, which drives their high learning rate and suboptimal choice behavior. First, since the models tested had only one means by which to generate a higher learning rate (the Kalman Filter determines learning rate based on estimated volatility and observation noise), and we know empirically that the asymptotic learning rate was higher for adolescents, I am not sure I buy this explanation of the provenance of the higher learning rate. At the very least it should be clarified that this was by design of the model -- another model, that explained higher asymptotic learning rates differently, might have suggest a different cause.

Second, the conclusions from modeling relied on fits of individual parameters. However, parameters of models, especially reinforcement learning models, are often difficult to recover reliably due to interdependence (e.g., a low learning rate and high inverse temperature can lead to the same likelihood as a high learning rate and low inverse temperature). In order to believe conclusions from parameter fits, I would first want to see some testing of the reliability of parameter recovery from the model. this can easily be done by simulating data from a known set of parameters, and then attempting to recover those parameters. Alternatively, or better, in addition, confusion matrixes across models can be plotted (that is, simulate data from each of the models, and fit it with all models to test how often the correct model is recovered). See Collins & Wilson's "Ten Simple Rules for the Computational Modeling of Behavioral Data" (https://psyarxiv.com/46mbn/) for more ideas on how to validate that you can make conclusions from model fits.

I had several other issues with the modeling: First, I was not sure what is the deviance information criterion (I have never encountered it before) -- how is this different from BIC? Are the results different if using the more standard BIC? Second, in Table 1 it was not clear that the difference between the DIC values of model 2B and 1B were significant -- what is the significance of a 1.5% difference between them, aggregated over subjects? It would be more compelling to see statistics, for instance, a t test on the difference in scores between the two models, across participants. I would also want to see a measure such as likelihood per trial, which would give some intuition regarding the goodness of fit of the models. This is especially important given the very low inverse temperature fits for the choice task -- such a high degree of randomness suggests a poor overall fit, given that the inverse temperature accounts not only for exploration, but also for all other sources of variance unexplained by the model. In Figure 5, the different Y axes in the adult graphs in A (top two) make it hard to read these graphs. Moreover, for S_1 I was not sure what the units were. Does the amount of initial variance that was fit by the model (in the order of hundreds) make sense?

Finally, the modeling results were, for the most part, described without statistical analysis. It is not clear to me that the difference in \\sigma^2 between the two age groups was less pronounced (line 387) and that the increase of the inverse temperature was steeper for older adults (line 400). All of these results would be more convincing with some statistical analysis, rather than just eyeballing the results and describing them.

On this last point, statistical analysis was also missing on line 212 -- please compare the learning rates in each age group to the adults if you would like to claim that the regulation of learning rates matures at age 15.

Minor comments:

- It was not clear to me until getting to the methods whether participants knew about the block structure. It would be good to make that clear upfront.

- Figure 3b - why are the bins different for the two age groups?

- Can the mediation analysis in Figure 6 be done on the reported uncertainty data as well?

- Will the data and code be made publicly available upon publication?

**Have all data underlying the figures and results presented in the manuscript been provided?**

Reviewer #1: No:

Reviewer #2: No: I may have missed it, but it was not clear to me whether data and code were to be made public upon publication

PLOS authors have the option to publish the peer review history of their article (what does this mean?). If published, this will include your full peer review and any attached files.

Reviewer #1: No

Reviewer #2: No

---

## [Decision Letter · Decision Letter 1]

26 Jul 2020

Dear Dr. Jepma,

Thank you very much for submitting your manuscript "Uncertainty-driven regulation of learning and exploration in adolescents: A computational account" for consideration at PLOS Computational Biology. As with all papers reviewed by the journal, your manuscript was reviewed by members of the editorial board and by several independent reviewers. The reviewers appreciated the attention to an important topic. Based on the reviews, we are likely to accept this manuscript for publication, providing that you modify the manuscript according to the review recommendations.

***

As you will see, reviewers are rather satisfied with your modifications. I simply want to highlight an important comment that was raised by reviewer #2: namely, that the softmax temperature is NOT a measure of people's tendency to explore. The softmax temperature also (if not mostly?) measures the inability of the model to capture people's choices, which may be driven by processes that are beyond the model's explanatory power (cf. model residuals). I suggest that, each time you interpret the softmax temperature in terms of exploration, you also explicitly recall the other (less interesting, but still important) interpretation.

***

Sincerely,

Jean Daunizeau

Associate Editor

PLOS Computational Biology

Natalia Komarova

Deputy Editor

PLOS Computational Biology

[LINK]

Reviewer's Responses to Questions

**Comments to the Authors:**

Reviewer #1: I found very intriguing that the model 2B (asymmetric reinforcement-learning model + dynamic softmax) performed best for the adolescents. I could find the information about the obtained values of alpha(+) and alpha(-). As there is some degree of disagreement in the literature (with Gershman 2015 reporting a negativity bias, Lefebvre 2017 reporting a positivity bias and Van Den Bos 2015 reporting both - as a function of the age) it would be very useful and informative for people in the field to report the results of the positive and negative learning rates of the model 2B as a function of the age group.

Other than that, I am satisfied by the revisions.

Reviewer #2: I thank the authors for their careful attention to all my comments, and I hope these were helpful. I am also extremely sorry for the delay with sending in this re-review — COVID-19 sent us all for a tailspin! In any case, I have only two remaining concerns, and they are mainly expositional:

The first concern is the unqualified over-interpretation of the meaning of the softmax inverse temperature parameter, as exemplified in the below sentences:

"The degree of choice randomness, or exploration, is often controlled by the inverse-temperature parameter, such that a higher inverse temperature results in a stronger tendency to choose the option with the highest expected value (i.e., less exploration)."

"Importantly, all learning models performed better when combined with a dynamic than a constant softmax function, for both age groups, suggesting that both the adults and adolescents did adjust their degree of exploration over time.”

It is important to remember that the softmax function is not necessarily the way the brain controls exploration — but rather how we link values in our model to probability of choice. This means that all misspecification of a model (that is, if the model learns values that differ, for any reason, from what is driving the participants’ behavior) will be folded into the softmax inverse temperature. If a model learns values that are not in accord with what is driving choice, the only way the model can fit the data is by “ignoring” these values to a larger extent, by lowering the inverse temperature. So while softmax can account for true exploration, we cannot assume that everything it accounts for is exploration.

The first sentence quoted above, in the introduction, conflates what the model does with what the subject does — the degree of exploration of the subject may be controlled through random and/or directed exploration, which are not accurately modeled or estimated by the softmax temperature (see, for example, Wilson, R. C., Geana, A., White, J. M., Ludvig, E. A., & Cohen, J. D. (2014). Humans use directed and random exploration to solve the explore–exploit dilemma. Journal of Experimental Psychology: General, 143(6), 2074, and other related papers from the lab of Robert Wilson). It is important to separate generative aspects of behavior (which we can test, but can’t assume) and what your model does.

The second sentence quoted above over-interprets the finding that a dynamic softmax function was a better fit for the data — this could also be due to the model being a worse explanation for the learning phase (the values it learns are different from those subjects learn, hence low inverse temperature) and a better explanation for the asymptotic phase (at that point, the learned values in the model and the learned values of subjects have converged (through different processes, perhaps) to the same values, so the model provides a better fit and the softmax inverse temperature can be increased. Basically, if a model and a human learn in different ways but arrive at the same conclusion (that is, they both learn to play the task), the best fit for a softmax model will be lower inverse temperature earlier on, and higher inverse temperature later on. Unfortunately, this may say nothing about the participants’ exploration strategy. I think it is important to be upfront about this and not misinterpret softmax findings as expressing conclusively something about participants’ strategy. Indeed, it *could be* that participants are reducing exploration over time, but we cannot know this without testing exploration directly, ideally with a tailored task, or, as you do, with some model-free analysis, for instance, looking at stay probability after a rewarded and unrewarded choice at different phases of the task. Given your model-free analysis, this is mainly an expositional point — I have seen so many papers confuse readers who are not versed with model fitting, to think that softmax = exploration. That is just not true.

My second concern is with this addition: "We compared the learning rates in each adolescent age group to the adults, and added the results on p. 11” — the missing statistical analysis here is the interaction between age and learning rate/learning rate reduction. Differences in p-values in of themselves are not always significant, so the most correct way to do this analysis is to test formally for an interaction with age, as you did for your other analyses. Only if the interaction is significant can we interpret the results you now describe in lines 218-231.

A few more minor comments:

- Although you say in the response letter that you now mention upfront that participants knew when a block changed in the first experiment, I did not find this in the experimental design section (around line 122 and later). Saying that the participants were informed about the task structure in advance does not imply that the block changes were signaled within the task itself. Please add.

- It is also not clear from the text or from Figure 1 how many blocks were in the choice task (line 137 suggests more than one — "the means differed 10 or 20 points, in half of the blocks each"), and whether subjects were informed of a block change (that is, are the subjects learning about new options from scratch several times? Or only once?)

- in line 304 you label the Kalman Filter “Model 2” but you already have another Model 2. You probably also meant to label the Pearce-Hall model “Model 4”.

- line 380 — I still wish there were some way to express the model fit in terms of (on average) how likely is each choice under the best fitting model (and each of the others). I find this very intuitive to understand (as we know that for a choice among two options, 50% is chance etc.). I know this is not trivial for MCMC fitting, but on the other hand, you extract

- line 395 "reducing their learning rates over time as a function of their estimation certainty.” — as far as I understand, the models did not model the reported estimation certainty, and did not show that the reduction of learning rate was a function of that. This statement is therefore confusing/misleading.

- the figures as uploaded were extremely grainy and rasterized and I could barely read them.

**Have all data underlying the figures and results presented in the manuscript been provided?**

Reviewer #1: **No: **I could not find any link where to download the data

Reviewer #2: Yes

PLOS authors have the option to publish the peer review history of their article (what does this mean?). If published, this will include your full peer review and any attached files.

Reviewer #1: No

Reviewer #2: No
---

## [Editor Report · Decision Letter 2]

20 Aug 2020

Dear Dr. Jepma,

We are pleased to inform you that your manuscript 'Uncertainty-driven regulation of learning and exploration in adolescents: A computational account' has been provisionally accepted for publication in PLOS Computational Biology.

Best regards,

Jean Daunizeau

Associate Editor

PLOS Computational Biology

Natalia Komarova

Deputy Editor

PLOS Computational Biology

---

## [Editor Report · Acceptance letter]

25 Sep 2020

PCOMPBIOL-D-19-01704R2 

Uncertainty-driven regulation of learning and exploration in adolescents: A computational account

Dear Dr Jepma,

I am pleased to inform you that your manuscript has been formally accepted for publication in PLOS Computational Biology. Your manuscript is now with our production department and you will be notified of the publication date in due course.

With kind regards,

Matt Lyles
